**Data Availability Statement:** All relevant data are within the paper and its Supporting Information files.

# Usability of mental illness simulation involving scenarios with patients with schizophrenia via immersive virtual reality: A mixed methods study

**Youngho Lee[1], Sun Kyung Kim[2,3]\*, Mi-Ran Eom[4]**

**1** Department of Computer Engineering, Mokpo National University, Muan-gun, Jeonnam, Korea,
**2** Department of Nursing, Mokpo National University, Muan-gun, Jeonnam, Korea, **3** Department of Biomedicine, Health & Life Convergence Sciences, Mokpo National University, Muan-gun, Jeonnam, Korea,
**4** Department of Nursing, Mokpo National University, Muan-gun, Jeonnam, Korea

\* skkim@mokpo.ac.kr

## Abstract

### Objectives

Schizophrenia is one of the most prevalent mental illnesses contributing to national burden worldwide. It is well known that mental health nursing education, including clinical placement, is still insufficient to reach the optimal level of competency in nursing students. This study suggests a new form of mental health virtual reality (VR) simulation that is user-friendly and engaging to improve education about schizophrenia, thereby improving its treatment.

### Method

A mixed-methods study was conducted with a total of 60 nursing students, using 360-degree videos of five different scenarios reflecting clinical symptoms of schizophrenia patients and related treatment tasks delivered via head-mounted displays (HMDs). We used a 17-item quantitative questionnaire and a 7-item open-ended qualitative questionnaire to evaluate the ease of use and usefulness of the VR simulation program and to identify areas where further improvement is required.

### Results

The VR simulation program was perceived as useful and exciting. Participants stressed that the high realism of the simulation increased their engagement in and motivation to learn about mental health nursing. Some participants made suggestions, such as further refining the picture and sound quality in order to achieve satisfactory educational outcomes.

### Conclusion

VR simulation using 360-degree videos and HMDs could serve as an effective alternative form of clinical training in mental health nursing. Education could be enhanced by its benefits of being engaging and exciting, as reported by this study's participants.

**Funding:** This work was supported by the National Research Foundation of Korea(NRF) grant funded by the Korea government(MSIT) (No. NRF-2019R1G1A1006737).

# Introduction

Schizophrenia is one of the mental illnesses contributing to national burden worldwide [1]. Given the complexity of this illness [2], nurses in psychiatric units require sufficient training and case experience; a lack of training may compromise the quality of care and result in negative outcomes for both the patients and the nurses. Most clinical placements for mental health nursing are currently limited to observation to ensure the safety of both patients and students [3], which does not produce a sufficient level of competency for quality nursing care.

Psychiatric nursing has a low profile among undergraduate nursing students and has been reported to be one of the least desirable career choices in the field [4, 5]. This lack of interest in mental health nursing has been attributed to anxiety related to working with patients with mental illness and feeling insufficiently prepared to perform this type of clinical work [6]. Thus, a lack of first-hand experience with mental health patients and unfamiliarity with mental disorders may result in stigmatization or fear among nursing students. Simulation has been evidenced as an effective alternative that could enhance clinical experience and maximize learning outcomes in mental health nursing [7–9]. Therefore, using integration strategies to create a rich simulation environment where students find learning interesting could effectively reduce barriers to engaging in mental health nursing practice.

## Virtual reality simulation

Technological development has benefited nursing education, as evidenced by the fact that educators have adopted various new approaches using virtual reality (VR) simulation. By using VR platforms, students are able to experience virtual worlds where it is possible to interact and communicate with virtual patients with a high sense of presence [10]. A recent review identified the advantages of VR simulation education as resulting in equal to better program outcomes compared to traditional learning methods and providing low-cost, space-efficient, and self-directed learning opportunities [11]. Furthermore, previous research in the field of mental health nursing education has found good applicability of VR in educational simulations and well-fostered emotional connections between learners and virtual patients [12].

Due to its strong potential for stimulating interactivity [13], VR has been used for simulation and skills training, and its effectiveness has been well established [14, 15]. Previous studies used virtual clinical simulation technologies that included a diverse range of situations, from simple scenarios such as medicine administration to more intense scenarios such as post-operative nursing care [9]. These VR simulations mostly used Internet-based virtual worlds, such as Second Life or CliniSpace, with non-immersive, desktop-based VR; however, these strategies could not create an optimal learning environment [9].

## Virtual reality simulation with new technologies

Several factors have been identified as benefits of VR that could be further enhanced with devices such as head-mounted displays (HMDs). First, the provision of sensory immersion has the advantage of disconnecting users from the outside world to reduce distractions and keep their minds from wandering [16, 17]. Second, place illusion leads to a better learning experience because users feel a sense of presence within the constructed virtual world [18]. Third, the emotional experience created in immersive VR could impact learning, as previous studies have found that VR simulation could be a more persuasive motivator [19, 20].

Previously considered high-cost devices, HMDs have become popular with the distribution of cost-effective products. HMDs elicit a sense of actually being present in a simulated world, which is hard to achieve via computer monitors or screens [21, 22]. Along with HMDs, the

educational advantages of using 360-degree videos have been reported [23–25]. Moreover, 360-degree videos incorporate recordings from every possible angle, which provides viewers with an experience similar to exploring the real world via panning or tilting the HMD [23].

In terms of mental health simulation, the most important learning objectives are for students to identify clinical symptoms and learn how to manage problematic symptoms of mentally ill patients. These symptoms are mostly subtle and demand careful observation of patients' facial expressions, speech, and behavior. For example, patients with advanced schizophrenia often lack facial expressions, due to symptoms associated with a diminished capacity for emotional expression [26]. This can cause difficulties for inexperienced nurses or nursing students in detecting changes in these patients' clinical condition. Thus, good acting that meticulously imitates these symptoms is essential for simulation education. Traditionally, standardized patients (SPs) were hired for mental health nursing simulations to portray patients with mentally illness. However, concerns have remained regarding the considerably high cost and time demands, and frequently changing SPs can exhaust faculties' resources for repeated training [7–9].

Providing scenarios using 360-degree videos and professional actors would be one possible solution for the problems above, due to the good quality of the actors' performance and sustainability of the program. Compared with artificial and avatar-based animation, 360-degree videos enable users to see a natural representation of mentally ill patients and clinical settings [27]. A further advantage is that scenarios can be quickly constructed due to the instant conversion of video recordings into VR that can be achieved using HMDs. Moreover, 360-degree videos with an immersive audiovisual experience using HMDs would help students retain acquired mental health nursing knowledge when they have no opportunities to access patients in person.

Using new technologies, immersive VR simulation can construct realistic learning situations that can improve educational outcomes. The objective of this study was to evaluate the usefulness of VR simulation for mental health nursing education using videos recorded by 360-degree cameras and delivered via HMDs. We hypothesized that a simulation experience that includes real people in clinical settings would induce active engagement of students with high learning satisfaction. We also hypothesized that a low amount of distraction from the outside world due to using HMDs would effectively evoke an emotional response from nursing students. A usability test of the VR simulation was conducted, and the potential for VR to be integrated into current mental health nursing education was examined.

## Methods

### Development

The scenarios were developed based on clinical situations in which registered nurses witnessed the problematic behavior of patients who were admitted to the psychiatric ward in an acute setting. Reflecting schizophrenia symptoms and hospital environments in acute settings, the scripts were written by a researcher and reviewed by a team of experts (two professors and one mental health nursing professional) and then edited regarding the realism and length of videos. Professional actors played in the scenarios that were recorded using 360-degree cameras. The scenarios were first rehearsed and then filmed incorporating feedback from experts regarding aspects such as facial expressions and interactions between actors. The final video clips were validated in terms of realism by two mental health nursing professionals and one nurse practitioner working in a psychiatric unit. The research team developed questions as tasks at several decision points to test participants' knowledge and capability for optimal

decision-making and critical thinking. The questionnaires were also evaluated and reviewed by a group of experts in terms of clinical relatedness and level of difficulty.

## Equipment and software

This study utilized an Oculus Go, which is a three degrees of freedom immersive HMD that does not need to be connected to a computer. The device is powered by Qualcomm's Snapdragon 821, uses a 2560x1440 resolution LCD, and has one controller as an input device. The 360-degree videos were recorded using Gear 360, which utilizes two fisheye lenses. The video resolution was 4096x2160 (24 frames per second) and used H.265 codex. Our software was developed with Unity3D, an engine for developing games and VR programs. When a participant ran the developed software, instructions were shown that briefly explained how to use it. Participants could then watch the 360-degree videos.

## Educational content

The educational objective of this study was to develop a program that could increase mental health nursing students' treatment competence for patients with schizophrenia in clinical settings. Five different scenarios were developed that incorporated five different symptoms of schizophrenia—risk of violation, auditory hallucination, visual hallucination, delusion, and risk of suicide. The individual scenarios conveyed clinical situations which required the students to attempt to choose an adequate reaction or response. In the risk of violation scenario, it was important for the users to identify and clear harmful objects within the surroundings, as patients could easily harm other people. The goal was to detect possible weapons and come up with strategies to calm patients down within a safe environment. In the risk of suicide scenario, close observation of patients' facial expression and behavior was necessary for the students to be able to identify related symptoms, as they were very subtle. Scenarios with patients experiencing hallucinations (auditory and visual) and delusions were designed to foster therapeutic communication that students need to determine symptoms based on patients' behavior and dialogue first, and then attempt therapeutic communication for patients to return to reality and align the treatment with trust in health professionals. At the end of each scenario, participants were given a scenario-based quiz and provided with vocalized communication with the virtual patient.

Two rooms were constructed: one patient room and one day room (a common use space where patients spend time during the day engaging in various activities such as watching tv or simple exercise). Professional actors performed according to the developed scenarios. After rehearsal, some revisions to the script were made to increase the intensity of the situation by actors and attended experts.

## Intervention

Prior to the VR simulation, detailed instructions regarding how to use the HMD and controller were provided in three phases that took about 15 to 20 minutes in total. First, a step-by-step introduction on where to look and which buttons to press was given using a video clip that appeared on the screen once the participant put on the HMD. Next, a research assistant demonstrated how to handle the controller while explaining the sequence of the VR simulation using pictures and text. Lastly, participants experienced the initial stage of the simulation program using devices. Manuals containing information on device features were also placed in the simulation rooms.

The VR intervention included a total of ten video clips: two video clips for each of the five scenarios of violence, auditory hallucination, visual hallucination, paranoia and drug refusal, that each ranged from 60 to 90 seconds in length. The video clips were delivered on the HMD

Oculus Go that allowed 360-degree videos to appear on the lens without connection to any other device (Fig 1). Prior to putting on the HMD, participants were first asked to choose a room. As participants entered each room, they were instructed to select numbers that appeared on the lens where the video was filmed in order to increase their sense of presence. No other hints were given such that participants were unaware of what would be happening in the selected VR world. After the end of each video clip, tasks related to skills such as appropriate decision-making (e.g., recognizing a threatening situation, choosing appropriate medication, and providing effective therapeutic communication) were delivered via on-screen questions. At the end of each scenario, communication-related tasks were given, showing suggestions to make the best responses following the patients' last dialogue which required therapeutic communication with health professionals. The rooms were open for two to three hours per day under the supervision of a research assistant, and participants were allowed to use the rooms several times with the devices available. Five HMDs were distributed to each group of seven to eight participants, and each room was occupied by three to four participants (Fig 2). Participants were instructed not to use the device for more than 20 minutes at a time and were guided to practice other nursing skills or engage in the theoretical study of schizophrenia.

## Usability evaluation

This study assessed the ease of use and usefulness of a VR simulation program using a 17-item quantitative rating scale questionnaire and a 7-item open-ended qualitative questionnaire. A usability test was conducted using a measure with items that were developed based on the existing literature and previously-developed usability tests for VR programs [10, 12]. This measure included a 17-item quantitative questionnaire and a 7-item qualitative questionnaire. The 17-item questionnaire allowed for responses on a scale ranging from 0 to 10 and consisted of two domains: ease of use (11 items) and usefulness (6 items). Except for one item (#7: perceived degree of difficulty in operating devices), a higher score meant a more favorable evaluation of the VR program. In the present study, Cronbach's alpha ($\alpha$) was .911.

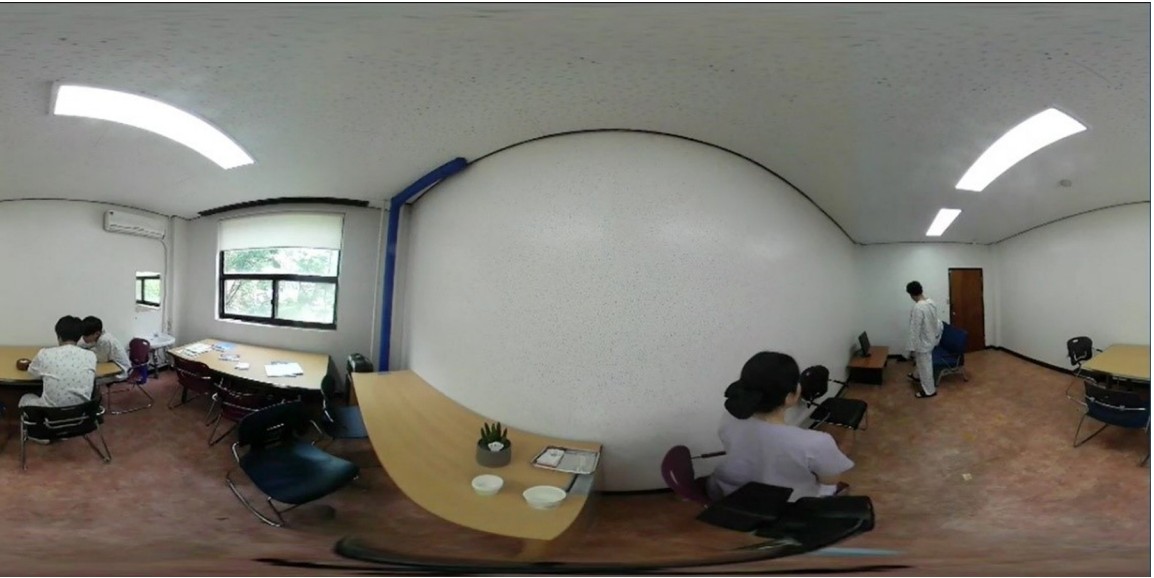

**Fig 1. A screenshot of our 360-degree video while filming a scenario of schizophrenia patients' symptoms.**

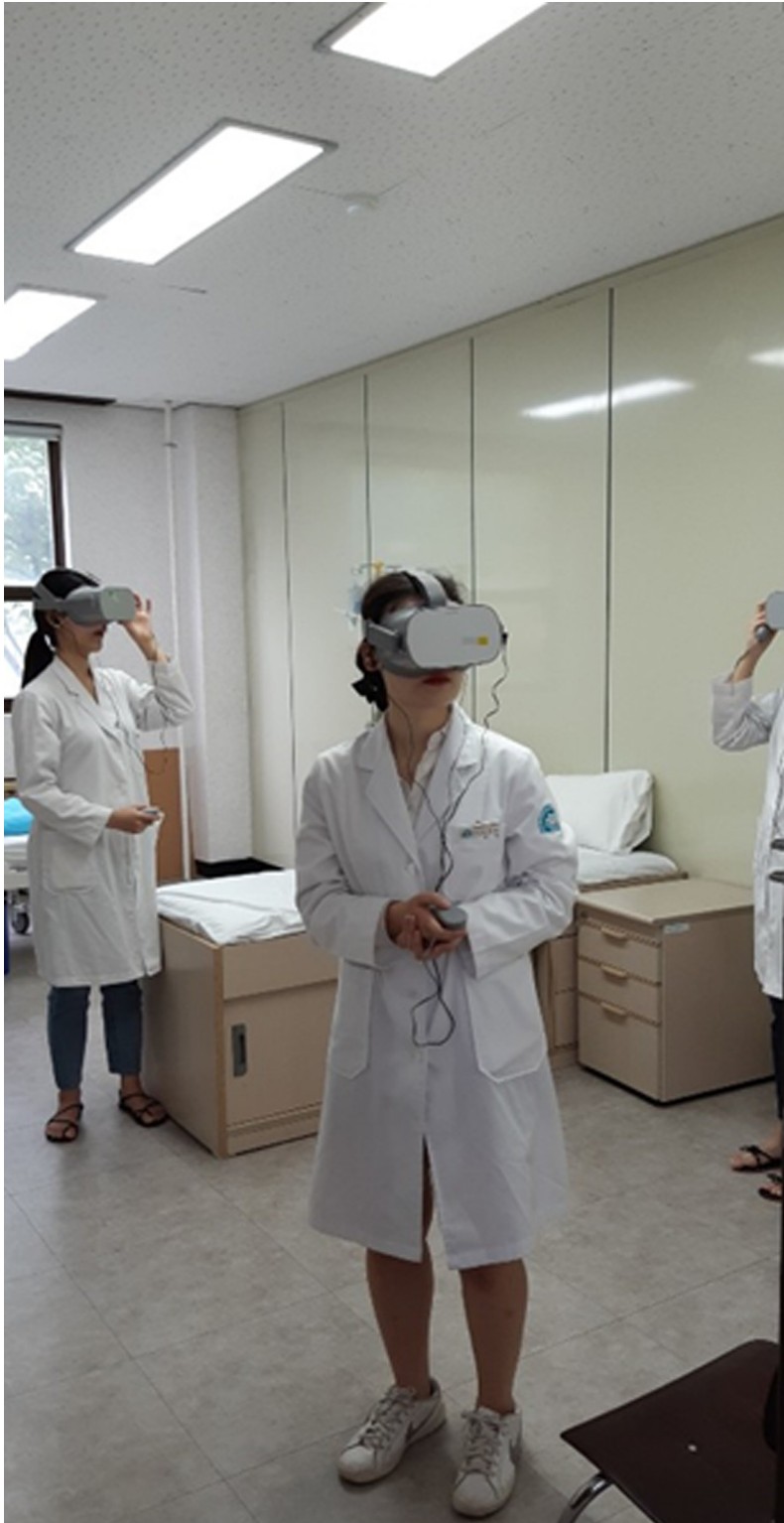

**Fig 2. Photos of the participants experiencing VR simulation using HMD.**

The open-ended qualitative questionnaire included 7 items that were asked to achieve an understanding of the advantages and disadvantages of the VR simulation program: 1) How did you find the program in general? 2) At the very beginning, was this program easy enough to use? Can you recommend any changes to improve its ease of use? 3) Was there any content that you found confusing during the program? 4) In terms of operating the devices, were there any difficulties you experienced? 5) How do you find this way of learning? Would it be helpful for your future clinical practice? 6) Repetitive use is important to achieve high effectiveness of educational programs. Can you recommend any changes to improve this program? 7) Do you have any other comments regarding this VR simulation program?

### Recruitment

Using convenience sampling, 60 nursing students were recruited from one university in Korea. All participants had completed most of their clinical placement trainings and were in their last semester of university.

### Data analysis

SPSS was used to analyze quantitative data from the usability survey and to calculate reliability. Qualitative responses were coded by a research assistant who reviewed and analyzed them to identify common themes.

### Ethical consideration

This study was conducted after receiving informed consent from all participants. The individual discussed in this manuscript has given written informed consent (as outlined in the PLOS consent form) to publish the case details herein. All procedures in this study were approved by Institutional Review Board of Mokpo National University in Korea (IRB No. MNUIRB-20190722-SB-005-01).

## Results

The study sample included 14 men (23.3%) and the mean age of participants was 23.6 years old. Academic achievement and perceived competency in communication varied widely among participants. One out of five students reported poor academic achievement and 16 participants (26.7%) reported a lack of competency in communication. Most participants did not have previous experience with VR (Table 1).

### Quantitative rating scale survey—usability

The highest mean score was recorded for the item "The initial education regarding devices and usage was appropriate" (9.25±1.00) within the ease of use domain, followed by the items "I am interested in using other VR simulation programs" (8.90±1.42) and "The VR simulation of schizophrenia program would be helpful for my future practice in clinical settings" (8.87 ±1.05). The mean score for perceived difficulties in device operation was 2.6 (Table 2).

### Qualitative essay questionnaires

**Overall experience of users.** The authenticity experienced through the VR simulation was the most frequently-reported comment in the qualitative questionnaire. Regarding their general perception of VR simulation, all participants had positive experiences and reported that it was realistic (n = 39, 65%) and an interesting experience (n = 15, 25%). Emotional

**Table 1. General characteristics of study participants (n = 60).**

| Characteristics | Categories | N(%) / M±SD |
|---|---|---|
| Gender | Male | 14 (23.3%) |
| | Female | 46 (76.7%) |
| Age | | 23.60±1.24 |
| Academic achievement | Good | 9 (15.0%) |
| | Fair | 39 (65.0%) |
| | Poor | 12 (20.0%) |
| Level of satisfaction with clinical placement of mental health nursing | High | 1 (1.7%) |
| | Moderate | 33 (55.0%) |
| | Low | 16 (26.7%) |
| Communication competency | High | 5 (8.3%) |
| | Moderate | 39 (65.0%) |
| | Low | 16 (26.7%) |
| Previous experience with VR | Yes | 4 (6.7%) |
| | No | 56 (93.3%) |

attachment to the virtual patients was evident from several participants' comments, such as: "I felt fearful when the actors behaved violently and came close to me."

**Experience using HMDs for 360-degree video.** Several participants reported discomfort using HMDs (n = 6), and made comments such as: "[The] HMD was not heavier than I expected; however, it soon caused unpleasant discomfort;" "I don't know why it is difficult to properly fix [the] HMD on my head;" and "The device seems not to be meant for people wearing glasses. It was painful." The 360-degree video experience was described as having met the educational goal of the mental health simulation well. Several participants (n = 13) reported feeling as if they were in the scenario situation, as one student stated, "I actually felt like I was in the middle of the room with the patients."

**Table 2. Responses to usability scale statement (n = 60).**

| Categories | No | Item | Range | Mean | SD |
|---|---|---|---|---|---|
| Ease of use | 1 | The VR simulation of schizophrenia program was easy to use. | 5−10 | 8.38 | 1.26 |
| | 2 | The initial education regarding devices and usage was appropriate. | 6−10 | 9.25 | 1.00 |
| | 3 | I wanted to continue watching other scenes and learning with related questions. | 4−10 | 8.65 | 1.57 |
| | 4 | The presented text was easy to read. | 2−10 | 7.48 | 2.15 |
| | 5 | The presented text was easy to understand. | 2−10 | 7.75 | 1.5 |
| | 6 | In each phase of VR simulation, it was clear what to do next. | 6−10 | 8.60 | 1.30 |
| | 7 | It was difficult to operate the devices (HMD and hand controller). | 0−8 | 2.60 | 2.61 |
| | 8 | The VR simulation was close to a real clinical situation. | 3−10 | 8.23 | 1.62 |
| | 9 | The video had good quality with high resolution. | 5−10 | 6.83 | 1.98 |
| | 10 | The pace of the VR simulation program was good. | 5−10 | 8.47 | 1.26 |
| | 11 | The audio quality was good. | 4−10 | 8.15 | 1.76 |
| Usefulness | 12 | The VR simulation of schizophrenia program would be helpful for my future practice in clinical settings. | 7−10 | 8.87 | 1.05 |
| | 13 | The VR simulation of schizophrenia program would be helpful for improving my communication skills. | 4−10 | 8.27 | 1.69 |
| | 14 | Besides the clinical training curriculum, the VR simulation of schizophrenia program could also be helpful in nursing education. | 5−10 | 8.58 | 1.37 |
| | 15 | The VR simulation of schizophrenia program would be helpful for improving my knowledge. | 6−10 | 8.63 | 1.09 |
| | 16 | The VR simulation of schizophrenia program would be helpful for improving my clinical decision-making capabilities. | 4−10 | 8.53 | 1.38 |
| | 17 | I am interested in using other VR simulation programs. | 3−10 | 8.90 | 1.42 |

**Perceived benefits.**  All participants agreed that this VR simulation program would eventually benefit their future clinical practice. Several participants commented on the educational benefits of the VR simulation (e.g., "This program provides a better understanding of the symptom-related nursing process for patients with schizophrenia."). Other participants made comments such as: "I may never have the courage to face patients with such mental conditions during clinical placement." In addition, some participants appreciated the safe learning environment of this VR simulation, with one student stating, "Exposure to dangerous clinical situations without risk is a great benefit of this learning program. I felt safe enough to attempt communication with patients in this VR simulation."

**Perceived difficulties.**  Two-thirds of participants (n = 40, 66.7%) found no difficulties in operating the devices and one out of four participants (n = 15, 25%) made comments such as: "It was a little confusing at first. However, I got used to it right away." Some problems were observed that may have been technical issues. Two participants found some degree of challenge with the buttons, and four participants made complaints such as: "The buttons did not work, even if I followed the manual." Several participants complained of low resolution (n = 17) and irregular audio quality (n = 9), saying that slight interruptions occurred during the simulation. Three participants reported dizziness after the VR simulation and five participants reported dizziness after extended use of HMD and raised the issue of safety, with one student commenting, "It is necessary to warn users not to use [the HMD] longer than 10–15 minutes."

**Recommendation from users.**  Many participants indicated that they would like to experience more diverse clinical situations that they would rarely observe during clinical placement as nursing students (n = 17, 28.3%). Others recommended the development of additional content to allow users to know the consequences of their decisions (n = 2). Several participants (n = 6) also expressed expectations for more complicated VR simulations, with comments such as: "It would be more interesting and helpful for students if we could actually communicate with virtual patients." A need for feedback regarding tasks was also revealed by some participants' (n = 5) requests for the correct answers with detailed explanations.

## Discussion

In this study, a usability test was conducted to determine the ease of use, usefulness, and perceived quality of a VR program using 360-degree video to simulate scenarios of patients with schizophrenia. To our knowledge, this was the first attempt to investigate the potential of VR technology for nursing education using simulated clinical situations with patients with schizophrenia in psychiatric wards.

In this study, participants found the VR simulation easy to use, engaging, and exciting. Most participants reported that they believed the content of this study's VR simulation was highly relevant for their education. Some participants expressed their desire to use this program with longer videos filmed of diverse clinical situations to better prepare themselves for in-person clinical training in psychiatric wards. Previous studies of simulation in mental health nursing have found evidence of improving clinical knowledge and reducing clinical placement anxiety [28]. In terms of training for effective communication, previous methodologies such as high-fidelity simulators have been found to provide relatively fewer beneficial effects due to the characteristics of patients with mental illnesses [29]. Suggestions could be made to provide VR training before students participate in simulations. Knowing that students perceive VR as exciting, the reduced stigma and anxiety that results from the use of VR could produce greater learning outcomes for clinical practice. As an alternative clinical training method, VR technology could effectively increase nursing students' confidence in their interactions with psychiatric patients.

Several technical issues were revealed in the 360-degree video VR simulation within the areas of functionality and human-computer interaction, specifically regarding low video resolution and poor text readability. Additionally, text information presented on-screen was reported to be unsatisfactory. Despite numerous attempts to provide clear written messages in VR, reported difficulties in displaying on-screen text have persisted [30, 31]. Nevertheless, a user-friendly software platform that includes elements that engage learners would be essential to eventually enhance motivation, knowledge retention, and critical-thinking skills. Verkuyl et al. [32] previously highlighted the importance of including a web designer who could easily build a user-friendly and customized VR environment and adopt features that learners will want to perform. Continuous design upgrades that reflect participant feedback would ensure good program usability.

Although the video resolution was unsatisfactory, the perceived realism of the program was good. The majority of participants reported that they experienced a good sense of immersion and presence during the VR simulation. One strategy used in the present study to enhance realism was the inclusion of professional actors instead of avatars. Our results align with those of a previous study that found that life-like experiences were created by actors and filming in places that resembled actual clinical settings [12]. The sense of realism experienced during this simulation could also be due to the participants' familiarization with the virtual world via a tour of the rooms where the videos were filmed and an introduction of the scenes prior to the VR simulation. However, no research has thus far been conducted to support this theory. Future research is therefore suggested to investigate whether viewing the actual filming space increases perceived realism for VR users.

The findings of this study showed that scores of perceived benefits for communication skills were relatively lower than scores of perceived usefulness for future practice. One of the limitations of this study is that the current version of the VR simulation used only a simple strategy for communication skills, prompting students to vocalize the best possible response to the last dialogue of virtual patients in the 360-degree video. The currently lower perceived utility for improving communication was also reflected by the fact that a number of users recommended the development of additional content so that they could actually practice communication with virtual patients. Based on the fact that previous studies have revealed the effectiveness of nursing simulations [33, 34], it may be useful to incorporate other devices to record communication between virtual patients and students. Given the opportunity to listen to the content of their own communication, learners are more likely to identify flaws and therefore improve their skills and confidence in therapeutic communication. Thorough discussion among experts of each discipline is required to select the most suitable and effective methods.

To be able to stimulate self-regulated learning, the VR simulation should focus on strategies to induce repetitive learning. In fact, participants requested feedback be incorporated into the VR simulation to receive detailed explanations on why their answers were right or wrong. A previous study [10] used the strategy of providing summarized information regarding participants' decisions, and participants showed appreciation and expressed its usefulness; however, detailed feedback could cause some concerns, as students may be less likely to conduct another round and too much text could cause tiredness in users. However, providing simple feedback, such as individual scores, could efficiently motivate students' self-directed learning. Additionally, Cooper and colleagues [35] suggested gaming elements be incorporated, yet the current version could not incorporate any. A simple strategy such as instant feedback showing scores or whether a user exceeded the set standard score could effectively induce motivation for repeated use.

An additional limitation of this study is that participating nursing students had previously completed their clinical placements. This suggests they may have already been exposed to

patients with mental disorders, which may have impacted their ratings of the experience. However, having a well-developed VR simulation with durability and broad application could be a strength. The robust VR simulation program could potentially benefit diverse populations, from inexperienced students to health professionals who need a refresher course.

## Conclusion

VR simulations have been considered useful learning strategies that could be effective alternatives to previously used nursing simulations. The present study conducted a usability test to explore users' experiences in terms of ease of use and perceived usefulness and collect constructive feedback from users. The perceived usability scores obtained in this study suggest that nursing students have the capacity to use VR technologies for educational purposes. In addition, users engaged well in the VR simulation and found this form of education to be exciting and useful. The findings indicate that VR-simulated scenarios of patients with schizophrenia involving 360-degree videos and HMDs have considerable potential to improve treatment competency among mental health nursing students.

## Supporting information

**S1 Data.**
(SAV)

**S2 Data.**
(XLSX)

**S3 Data.**
(PDF)

## Author Contributions

**Conceptualization:** Youngho Lee, Sun Kyung Kim, Mi-Ran Eom.

**Data curation:** Sun Kyung Kim.

**Formal analysis:** Sun Kyung Kim.

**Funding acquisition:** Sun Kyung Kim.

**Investigation:** Sun Kyung Kim.

**Methodology:** Sun Kyung Kim.

**Project administration:** Youngho Lee, Sun Kyung Kim.

**Resources:** Sun Kyung Kim.

**Software:** Youngho Lee, Sun Kyung Kim.

**Supervision:** Youngho Lee, Sun Kyung Kim, Mi-Ran Eom.

**Validation:** Sun Kyung Kim.

**Visualization:** Sun Kyung Kim.

**Writing – original draft:** Youngho Lee, Sun Kyung Kim.

**Writing – review & editing:** Youngho Lee, Sun Kyung Kim, Mi-Ran Eom.

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
