## [Decision Letter · Decision Letter 0]

12 Jun 2020

PONE-D-19-34647

Usability of mental illness simulation involving scenarios with patients with schizophrenia via immersive virtual reality: a mixed methods study

PLOS ONE

Dear Dr. Kim,

Thank you for submitting your manuscript to PLOS ONE. After careful consideration, we feel that it has merit but does not fully meet PLOS ONE’s publication criteria as it currently stands. Therefore, we invite you to submit a revised version of the manuscript that addresses the points raised during the review process.

We look forward to receiving your revised manuscript.

Kind regards,

Frédéric Denis, Ph.D.

Academic Editor

PLOS ONE

Journal Requirements:

2. Please include your tables as part of your main manuscript and remove the individual files. Please note that supplementary tables (should remain/ be uploaded) as separate "supporting information" files

3. We note that Figures 2 and 3 includes an image of a [patient / participant / in the study]. 

Additional Editor Comments (if provided):

Reviewers' comments:

Reviewer's Responses to Questions

**Comments to the Author**

1. Is the manuscript technically sound, and do the data support the conclusions?

Reviewer #1: Partly

Reviewer #2: Yes

2. Has the statistical analysis been performed appropriately and rigorously? 

Reviewer #1: No

Reviewer #2: Yes

3. Have the authors made all data underlying the findings in their manuscript fully available?

Reviewer #1: No

Reviewer #2: Yes

4. Is the manuscript presented in an intelligible fashion and written in standard English?

Reviewer #1: Yes

Reviewer #2: Yes

5. Review Comments to the Author

Reviewer #1: 1. This paper evaluated a VR simulation for its usability in effectiveness and usefulness. One thing is unclear in their study. Are participants explicitly asked to consider the traditional education approach as the baseline? Are participants comparatively comparing against the baseline, or are they evaluating the VR simulation independently? If it is the former case, please state that explicit instructions were given to compare against the traditional approach. If it is the latter case, the authors should provide more details on the results of 7-item open-ended qualitative questionnaire.

2. One shortcoming of this paper is that it lacks explanation of why 360 degree video was chosen as the medium and what were their effects in their VR simulation? Is 360-degree video beneficial for providing scenarios with schizophrenia patients? If so, I would suggest the authors to explicitly mention those motivations. For example, (I'm just providing a made-up example for brevity) 'it is important to avoid eye-contact with the patient in some cases, so 360 degree video could effectively simulate that'. I did not find any direct questionnaire items related to 360-degree video, so this part could be improved with further explanations.

3. For the developed educational contents, are there any interactivity implemented? For example, were participants able to trigger different behavior of the virtual patients (playing different video clips, perhaps)? Also the paper mentioned that there were scenario-based quizzes. Did you collect the scores of these quizzes and compared to other approaches (i.e., traditional textbook)? This part was unclear to me. Also, how one can improve their communication skills if there was no interaction/interactivity component in the VR simulation?

4. Some questionnaire items are misleading to me. I think "good acting part" by real actors and "VR simulation" elements should be evaluated separately. But current items have integrated both of these components in questionnaire items. For example, one can argue that 2D video with "good acting" is more effective than badly acted 360-degree video contents. As mentioned in point 2, I do not see direct questions evaluating use of HMD or 360-degree videos that are more related with motions and directions of virtual patients. Rather there are 2 questionnaire items with use of the text.

Reviewer #2: - This study proposed a method to simulate the treatment of schizophrenia

using VR and verified its effectiveness through usability testing.

- After shooting real patients and experts using 360 cameras rather than

simulations through graphics, VR content was created to provide realistic

training.

- In particular, for education on the treatment of schizophrenia, I think

that the VR-applied method is safe and can be used for long-term monitoring.

- It would be helpful to understand the paper if there are additional

explanations on the following.

---- Five scenarios are outlined, but I am curious about the differences in

the scenarios. Also, I wonder if there is a difference in usability

depending on the scenario.

---- Items 4, 5, and 7 in the questionnaire seem to have differences in

usability depending on the experience of using VR.

---- What does "one day room" mean in 160 lines?

---- I wonder what kind of interaction is used during education. It is

necessary to explain what kind of user input was sent through the wand to

make it interactive.

---- Considering that the treatment of schizophrenia does not work at once,

it is also necessary to test whether it is applicable to long-term

monitoring and education as a future work.

6. PLOS authors have the option to publish the peer review history of their article (what does this mean?). If published, this will include your full peer review and any attached files.

Reviewer #1: Yes: Hyoseok Yoon

Reviewer #2: Yes: Ahyoung Choi

---

## [Author Response · Author response to Decision Letter 0]

30 Jun 2020

Dear Editor and Reviewers,

Thank you very much for reviewing our paper and for your thoughtful comments. We have addressed them one by one in the table below. Please find the newly added or amended sections of the manuscript marked in red font.

Reviewer’s comment

Reviewer #1: 1. This paper evaluated a VR simulation for its usability in effectiveness and usefulness. One thing is unclear in their study. Are participants explicitly asked to consider the traditional education approach as the baseline? Are participants comparatively comparing against the baseline, or are they evaluating the VR simulation independently? If it is the former case, please state that explicit instructions were given to compare against the traditional approach. If it is the latter case, the authors should provide more details on the results of 7-item open-ended qualitative questionnaire. 

Author’s response 

It is the latter case and more details were added throughout the manuscript. We have further analyzed the responses to the 7 item open-ended qualitative questionnairs and reinforced the results and discussion sections.

2. One shortcoming of this paper is that it lacks explanation of why 360 degree video was chosen as the medium and what were their effects in their VR simulation? Is 360-degree video beneficial for providing scenarios with schizophrenia patients? If so, I would suggest the authors to explicitly mention those motivations. For example, (I'm just providing a made-up example for brevity) 'it is important to avoid eye-contact with the patient in some cases, so 360 degree video could effectively simulate that'. I did not find any direct questionnaire items related to 360-degree video, so this part could be improved with further explanations. 

Author’s response 

We added the following paragraph to further explain the recognized benefits of 360 degree video in this research. 

“In terms of the mental health simulation, the most important learning objectives are for students to identify clinical symptoms and learn how to manage problematic symptoms of mentally ill patients. These symptoms are mostly subtle, demanding careful observation of facial expressions, and the speech and behavior of patients. Thus, good acting that meticulously imitates these symptoms is essential. Traditionally, SPs were hired for mental health nursing simulation to reenact mentally ill patients. However, concerns remained regarding the considerably high cost and time demand associated with hiring SPs. The reason for this was that previously trained SPs were found not able to produce effective educational outcomes, and frequently changing SPs exhausted the faculties’ resources for repeated training [4-6]. The provision of scenarios using a 360 degree video and professional actors would be one possible solution for the problems above, due to the good quality of the actors’ performance and sustainability of the program. In addition, comparing with artificial and avatar-based animation, 360 degree videos enable users to see a natural representation of mentally ill patients as well as the clinical settings.” (P 6-7, L 111-124) 

3. For the developed educational contents, are there any interactivity implemented? For example, were participants able to trigger different behavior of the virtual patients (playing different video clips, perhaps)? Also the paper mentioned that there were scenario-based quizzes. Did you collect the scores of these quizzes and compared to other approaches (i.e., traditional textbook)? This part was unclear to me. Also, how one can improve their communication skills if there was no interaction/interactivity component in the VR simulation? 

Author’s response

Further details regarding the five scenarios concerning their purpose and educational goals were added. 

“The individual scenarios conveyed clinical situations which required the students to attempt to choose an adequate reaction or response. In the risk of violation scenario, it was important for the users to identify and clear harmful objects within surroundings as patients easily harm other people. The goal was to detect possible weapons and come up with strategies to calm patients down within a safe environment. In the risk of suicide scenario, close observation of patients' facial expression and behavior was necessary for the students to be able to identify related symptoms as they were very subtle. The patients experiencing hallucination (auditory and visual) and delusion scenarios were designed to foster therapeutic communication that students need to determine symptoms based on patients' behavior and dialogue first, and then attempt therapeutic communication for patients to return to reality and align the treatment with trust in health professionals.” (P7, L172-182).

We were not able to collect the scores of the quizzes as this study primarily served as a usability test, and thus we are aware that our program needs further upgrade to note the score the quizzes the students took.

We planned future studies with focus on examining the effects of VR simulation on knowledge and skill acquisition as well as learning satisfaction compared with traditional scenario-based simulation education.

In addition, this current version could not incorporate the features that ensure an optimal interaction between virtual patients and students. One thing that we tried within the current version was prompting students’ to verbal communication by displaying the message ‘please make the right response following the patients last dialogue’ 

Based on your valuable comments were added a limitation to the discussion stating that further improvement is required for future VR simulation.

“One of the limitations of this study is that the current version of the VR simulation used only a simple strategy for communication skills, prompting students to vocalize the best possible response to the last dialogue of virtual patients in the 360 degree video.” (P 18-19, L388-391)

“Yet, the current version could not incorporate a scoring system, a simple strategy such as instant feedback showing the score or whether they exceeded the set standard score would effectively induce motivation for repeated usage.” (P19, L411-414)

We also added the following sentence, providing a detailed description of the educational tasks embaded in the simulation program. 

“After the end of each video clip, tasks related to skills such as appropriate decision-making (e.g., recognizing a threatening situation, choosing appropriate medication, and providing effective therapeutic communication) were delivered via on-screen questions. At the end of each scenario, communication related tasks were given showing suggestions to make the best responses following the patients’ last dialogue which required therapeutic communication with health professionals.” (P11, L213-218)

4. Some questionnaire items are misleading to me. I think "good acting part" by real actors and "VR simulation" elements should be evaluated separately. But current items have integrated both of these components in questionnaire items. For example, one can argue that 2D video with "good acting" is more effective than badly acted 360-degree video contents. As mentioned in point 2, I do not see direct questions evaluating use of HMD or 360-degree videos that are more related with motions and directions of virtual patients. Rather there are 2 questionnaire items with use of the text. 

Author’s response

The main purpose of developing a VR simulation using a 360 degree video was to provide a real-life-like simulation within a safe environment. This simulation was created for students who have already completed clinical placements in psychiatric wards. As mentioned in the manuscript, one of the problems of this clinical placement is that students are allowed observation only. This has necessitated the use of a simulation where students can apply nursing interventions based on their clinical decision making. Our interest was whether a VR simulation using a 360 degree could well deliver a virtually constructed clinical scenario reflecting a real world experience. Thus, good acting was not somthing that needed evaluation as it had to be ensured for this educational purpose.

We added a more detailed evaluation obtained from the students regarding the HMD and 360 degree video.

Experience using HMD for 360 degree video

“Several participants reported discomfort using HMD (n=6), and made comments such as: ‘HMD was not heavier than I expected, however, it soon caused unpleasant discomfort which was aggravated with earphones’; ‘I don’t know why it is difficult to properly fix HMD on my head. It is somehow annoying adjusting the bands. I want to learn an easier and faster way to it put on’; ‘The device seems not to be meant for people wearing glasses. However I adjusted the HMD, and it was painful.’ The experience of the 360 degree video was said to have well met the educational goal of the mental health simulation. Several participants (n=13) reported feeling that they were in the scenario situation saying ‘I actually felt like I was in the middle of the room with the patients’; ‘I felt weird when I looked around the room in HMD, however It was a bit frustrating when I realized that I could not get closer or even move forward to the patients.’” (P17, L295-306) 

Reviewer #2: - This study proposed a method to simulate the treatment of schizophrenia using VR and verified its effectiveness through usability testing.

- After shooting real patients and experts using 360 cameras rather than simulations through graphics, VR content was created to provide realistic training.

- In particular, for education on the treatment of schizophrenia, I think that the VR-applied method is safe and can be used for long-term monitoring.

- It would be helpful to understand the paper if there are additional explanations on the following.

1. Five scenarios are outlined, but I am curious about the differences in the scenarios. Also, I wonder if there is a difference in usability depending on the scenario. 

Author’s response

We added more information about the scenarios in terms of their educational purpose and differences among them. 

“The individual scenarios conveyed clinical situations which required the students to attempt to choose an adequate reaction or response. In the risk of violation scenario, it was important for the users to identify and clear harmful objects within surroundings as patients easily harm other people. The goal was to detect possible weapons and come up with strategies to calm patients down within a safe environment. In the risk of suicide scenario, close observation of patients' facial expression and behavior was necessary for the students to be able to identify related symptoms as they were very subtle. The patients experiencing hallucination (auditory and visual) and delusion scenarios were designed to foster therapeutic communication that students need to determine symptoms based on patients' behavior and dialogue first, and then attempt therapeutic communication for patients to return to reality and align the treatment with trust in health professionals.” (P7, L172-182).

2. Items 4, 5, and 7 in the questionnaire seem to have differences in usability depending on the experience of using VR. 

Author’s response

The reviewers’ comment is reflected in the results section (P18, L324-327). and the discussion section. (P17, L357-361)

“Some problems were observed that may have been technical issues: two participants found some degree of challenge with the buttons and four other participants made complaints such as: “The buttons did not work even if I followed the manual”; ‘It was difficult to figure out the functions of the buttons at first, especially the symbols on the buttons seemed not to be related to their functions.’ One the other hand others reported ease of use saying ‘Oculus seems beneficial for educational use as it has simple design with buttons on a simple joystick.’”

(P15, L315-318).

“There was also a minor technical issue of the multiple and meaningless symbols on the buttons causing confusion of their functions. However, this difficulty seems to have been limited those who lacked previous experience with computer games and VR devices. In a previous article, Verkuyl [26] highlighted the importance of including a web designer who could easily build a user-friendly and customized environment and adopt features that learners will want to perform. A design continuous upgrade reflecting the participants’ feedback would ensure good usability of the programs.” (P20, L366-372)

3. What does "one day room" mean in 160 lines? 

Author’s response

It is a room for daytime activity.

“Two rooms were constructed: one patient room and one day room (a common use space where patients spend time during the day engaging in various activities such as watching tv or simple exercise).” (P10, L189-191) 

4. I wonder what kind of interaction is used during education. It is necessary to explain what kind of user input was sent through the wand to make it interactive. 

Author’s response

We provided more details regarding tasks given during the VR simulation. In terms of interaction, this is a limitation of this study as the current version could not well incorporate strategies that induce active interaction within the simulation. We now clearly stated this as a limitation in the manuscript.

“After the end of each video clip, tasks related to skills such as appropriate decision-making (e.g., recognizing a threatening situation, choosing appropriate medication, and providing effective therapeutic communication) were delivered via on-screen questions. At the end of each scenario, communication related tasks were given showing suggestions to make the best responses following the patients’ last dialogue which required therapeutic communication with health professionals.” (P11, L215-220)

“The findings of this study showed that scores of perceived benefits for communication skills were relatively lower than scores of perceived usefulness for future practice. One of the limitations of this study is that the current version of the VR simulation used only a simple strategy for communication skills, prompting students to vocalize the best possible response to the last dialogue of virtual patients in the 360 degree video. The currently lower perceived utility for improving communication was also reflected by the fact that a number of users recommended the development of additional content so that they could actually practice communication with virtual patients.” (P 21, L397-400)

5. Considering that the treatment of schizophrenia does not work at once, it is also necessary to test whether it is applicable to long-term monitoring and education as a future work. 

Author’s response

We have added the following paragraph to emphasize the beneficial effects of VR technology for education of health professionals. 

“Having a well-developed VR simulation with its durability and broad application are also the strengths of the present study. With the use of VR technology, the robust simulation program could benefit students in remote areas. In addition, continuous use can produce long-lasting effects for nurses’ education and then practice, acting as means of real-time monitoring and updating the quality of nursing care. In addition, the application of this simulation to other health professionals should be considered as it would effectively improve the understanding of patients with mental disorders.” (P22, L424-430)

---

## [Decision Letter · Decision Letter 1]

6 Aug 2020

PONE-D-19-34647R1

Usability of mental illness simulation involving scenarios with patients with schizophrenia via immersive virtual reality: a mixed methods study

PLOS ONE

Dear Dr. Kim,

Thank you for submitting your manuscript to PLOS ONE. After careful consideration, we feel that it has merit but does not fully meet PLOS ONE’s publication criteria as it currently stands. Therefore, we invite you to submit a revised version of the manuscript that addresses the points raised during the review process.

We look forward to receiving your revised manuscript.

Kind regards,

Frédéric Denis, Ph.D.

Academic Editor

PLOS ONE

Reviewers' comments:

Reviewer's Responses to Questions

**Comments to the Author**

1. If the authors have adequately addressed your comments raised in a previous round of review and you feel that this manuscript is now acceptable for publication, you may indicate that here to bypass the “Comments to the Author” section, enter your conflict of interest statement in the “Confidential to Editor” section, and submit your "Accept" recommendation.

Reviewer #1: All comments have been addressed

Reviewer #3: (No Response)

2. Is the manuscript technically sound, and do the data support the conclusions?

Reviewer #1: Yes

Reviewer #3: Yes

3. Has the statistical analysis been performed appropriately and rigorously? 

Reviewer #1: Yes

Reviewer #3: N/A

4. Have the authors made all data underlying the findings in their manuscript fully available?

Reviewer #1: Yes

Reviewer #3: (No Response)

5. Is the manuscript presented in an intelligible fashion and written in standard English?

Reviewer #1: Yes

Reviewer #3: Yes

6. Review Comments to the Author

Reviewer #1: (No Response)

Reviewer #3: This is a useful feasibility study of a VR platform for exposing nursing students to interactions with patients with serious mental illness on an inpatient unit. The study had several strengths. A VR mental health training platform could be useful and easily disseminated; not just for nursing but for other mental health disciplines. It was a significant undertaking to develop the content and programming to create the VR training platform. The number of participants (N=60) was also respectable. The authors also did a good job responding to prior reviewer concerns. A few concerns remain.

The authors frequently refer to the need for VR skilled actors to simulate the "subtle symptoms" of people with schizophrenia. It is not at all clear what this means. Does this refer to facial expressions that indicate hallucination experiences or paranoia? More could be said about this as it relates to the need for a VR platform. Perhaps the figure showing the manual/remotes (Fig 1) could be replaced with a figure illustrating what the authors mean by this, which would be much more helpful.

Related to this, the need for a VR training platform is not set up well in the introduction. The authors note in the abstract, discussion and elsewhere that the VR experience was "exciting," and comment on motivation to learn about mental health nursing. Safety of interacting on inpatient units is also noted as a barrier to training, as well as the work involved in repeatedly training simulation actors. Is motivation or interest in mental health nursing a barrier; is there a literature on this? How does VR overcome interest in this by being an "exciting" platform? Are students more competent if trained in VR first? The specific problems with nursing training, and how a VR platform solves these problems, is not clearly explained in the introduction, and this set up should drive a more concise discussion.

There is very little data in this paper, which is essentially a feasibility study introducing the novel VR training platform. Given this limited data and focus, the manuscript is very long. A more concise introduction laying out the problems and solutions of VR training as described above, and concise discussion of this and the results would improve the manuscript. For example, there is a lot of discussion of the readability of the text and details of the platform in the discussion which is not as helpful as a focus on the purpose and acceptability of the VR training platform.

The nursing students enrolled in the study had already completed their clinical placements. This suggests they may have already been exposed to patients with mental disorders, which may have impacted their ratings of the experience. If so, since the VR platform is directed at training new students, this limitation should be mentioned.

7. PLOS authors have the option to publish the peer review history of their article (what does this mean?). If published, this will include your full peer review and any attached files.

Reviewer #1: **Yes: **Hyoseok Yoon

Reviewer #3: No

---

## [Author Response · Author response to Decision Letter 1]

17 Aug 2020

Reviewer #3: This is a useful feasibility study of a VR platform for exposing nursing students to interactions with patients with serious mental illness on an inpatient unit. The study had several strengths. A VR mental health training platform could be useful and easily disseminated; not just for nursing but for other mental health disciplines. It was a significant undertaking to develop the content and programming to create the VR training platform. The number of participants (N=60) was also respectable. The authors also did a good job responding to prior reviewer concerns. A few concerns remain.

Authors response

We appreciate reviewers’ meticulous and constructive comments. We have made revisions accordingly.

1.The authors frequently refer to the need for VR skilled actors to simulate the "subtle symptoms" of people with schizophrenia. It is not at all clear what this means. Does this refer to facial expressions that indicate hallucination experiences or paranoia? More could be said about this as it relates to the need for a VR platform. Perhaps the figure showing the manual/remotes (Fig 1) could be replaced with a figure illustrating what the authors mean by this, which would be much more helpful.

Authors response

We have added a more detailed explanation regarding disease symptoms and related issues in nursing education.

Introduction

In terms of mental health simulation, the most important learning objectives are for students to identify clinical symptoms and learn how to manage problematic symptoms of mentally ill patients. These symptoms are mostly subtle and demand careful observation of patients’ facial expressions, speech, and behavior. For example, patients with advanced schizophrenia often lack facial expressions, due to symptoms associated with a diminished capacity for emotional expression [26]. This can cause difficulties for inexperienced nurses or nursing students in detecting changes in these patients’ clinical conditions. Thus, good acting that meticulously imitates these symptoms is essential for simulation education. Traditionally, standardized patients (SPs) were hired for mental health nursing simulation to portray patients with mentally illness. However, concerns have remained regarding the considerably high cost and time demands, and frequently changing SPs can exhaust faculties’ resources for repeated training [7-9]. 

2. Related to this, the need for a VR training platform is not set up well in the introduction. The authors note in the abstract, discussion and elsewhere that the VR experience was "exciting," and comment on motivation to learn about mental health nursing. Safety of interacting on inpatient units is also noted as a barrier to training, as well as the work involved in repeatedly training simulation actors. Is motivation or interest in mental health nursing a barrier; is there a literature on this? How does VR overcome interest in this by being an "exciting" platform? Are students more competent if trained in VR first? The specific problems with nursing training, and how a VR platform solves these problems, is not clearly explained in the introduction, and this set up should drive a more concise discussion.

Authors response

We added the following paragraphs to the Introduction and Discussion sections:

Introduction

Psychiatric nursing has a low profile among undergraduate nursing students and has been reported to be one of the least desirable career choices in the field [4,5]. This lack of interest in mental health nursing has been attributed to anxiety related to working with patients with mental illness and feeling insufficiently prepared to perform this type of clinical work [6]. Thus, a lack of first-hand experience with mental health patients and unfamiliarity with mental disorders may result in stigmatization or fear among nursing students. Simulation has been evidenced as an effective alternative that could enhance clinical experience and maximize learning outcomes in mental health nursing [7-9]. Therefore, using integration strategies to create a rich simulation environment where students find learning interesting could effectively reduce barriers to engaging in mental health nursing practice. 

Discussion

In this study, participants found the VR simulation easy to use, engaging, and exciting. Most participants reported that they believed the content of this study’s VR simulation was highly relevant for their education. Some participants expressed their desire to use this program with longer videos filmed of diverse clinical situations to better prepare themselves for in-person clinical training in psychiatric wards. Previous studies of simulation in mental health nursing have found evidence of improving clinical knowledge and reducing clinical placement anxiety [28]. In terms of training for effective communication, previous methodologies such as high-fidelity simulators have been found to provide relatively fewer beneficial effects due to the characteristics of patients with mental illnesses [29]. Suggestions could be made to provide VR training before students participate in simulations. Knowing that students perceive VR as exciting, the reduced stigma and anxiety that results from the use of VR could produce greater learning outcomes for clinical practice. As an alternative clinical training method, VR technology could effectively increase nursing students’ confidence in their interactions with psychiatric patients.

3. There is very little data in this paper, which is essentially a feasibility study introducing the novel VR training platform. Given this limited data and focus, the manuscript is very long. A more concise introduction laying out the problems and solutions of VR training as described above, and concise discussion of this and the results would improve the manuscript. For example, there is a lot of discussion of the readability of the text and details of the platform in the discussion which is not as helpful as a focus on the purpose and acceptability of the VR training platform.

Authors response

To reflect the reviewer’s comment, we eliminated redundant sentences and a figure, making the Introduction, Results, and Discussion sections more concise.

4. The nursing students enrolled in the study had already completed their clinical placements. This suggests they may have already been exposed to patients with mental disorders, which may have impacted their ratings of the experience. If so, since the VR platform is directed at training new students, this limitation should be mentioned.

Authors response

To address this issue, the following paragraph has been added as a study limitation:

Limitation

An additional limitation of this study is that participating nursing students had previously completed their clinical placements. This suggests they may have already been exposed to patients with mental disorders, which may have impacted their ratings of the experience. However, having a well-developed VR simulation with durability and broad application could be a strength. The robust VR simulation program could potentially benefit diverse populations, from inexperienced students to health professionals who need a refresher course.

---

## [Editor Report · Decision Letter 2]

18 Aug 2020

Usability of mental illness simulation involving scenarios with patients with schizophrenia via immersive virtual reality: a mixed methods study

PONE-D-19-34647R2

Dear Dr. Kim,

We’re pleased to inform you that your manuscript has been judged scientifically suitable for publication and will be formally accepted for publication once it meets all outstanding technical requirements.

Kind regards,

Frédéric Denis, Ph.D.

Academic Editor

PLOS ONE
---

## [Editor Report · Acceptance letter]

1 Sep 2020

PONE-D-19-34647R2 

Usability of mental illness simulation involving scenarios with patients with schizophrenia via immersive virtual reality: a mixed methods study 

Dear Dr. Kim:

I'm pleased to inform you that your manuscript has been deemed suitable for publication in PLOS ONE. Congratulations! Your manuscript is now with our production department. 

Kind regards, 

on behalf of

Dr. Frédéric Denis 

Academic Editor

PLOS ONE